# Tuning Autophagy for Improved Liver Transplant Outcomes: Insights from Experimental Models

**DOI:** 10.3390/biom15060797

**Published:** 2025-05-31

**Authors:** Mina Kolahdouzmohammadi, Graziano Oldani

**Affiliations:** 1Department of Surgery, Faculty of Medicine, University of British Columbia, Vancouver, BC V6T 1Z3, Canada; mina.kolahdouz@ubc.ca; 2BC Children’s Hospital Research Institute, Vancouver, BC V5Z 4H4, Canada

**Keywords:** liver transplantation, graft rejection, autophagy, xenotransplantation

## Abstract

Liver transplantation faces significant challenges, primarily due to the severe shortage of organs—aggravated by the increasing prevalence of liver diseases—and graft loss due to the consequences of ischemia/reperfusion injury (I/RI) and rejection. A recent study highlights the critical role of autophagy, a cellular breakdown and recycling mechanism, in addressing these issues. This article examines the role of autophagy in liver transplantation, focusing on organ preservation and recovery after surgery, as well as its potential to regulate immune responses and increase graft survival. Additionally, it will cover the role of autophagy in xenotransplantation, a prospective solution to the organ scarcity crisis. Ultimately, it assesses the importance of precisely timing autophagy modulation—whether induction or inhibition—to enhance transplantation outcomes, while identifying key knowledge gaps and future research directions.

## 1. Introduction

Liver disease has emerged as a leading cause of mortality in recent years, with approximately 25% of the global population being at risk due to factors such as viral infections, excessive alcohol consumption, and metabolic dysfunction-associated steatohepatitis (MASH) [1]. Currently, liver transplantation is the only effective treatment for end-stage liver disease and is performed globally. However, the rising prevalence of liver disease has significantly increased the need for transplantation, aggravating the existing urgent scarcity of donor organs [2].

In addition to organ shortage, liver transplantation has considerable obstacles, notably organ preservation, which is essential for graft survival. Researchers are diligently enhancing preservation methods to optimize transplant results [3,4]. Nevertheless, this is just the beginning of the story as following the completion of the surgery, additional complications arise.

Following reperfusion of the transplanted liver, ischemia/reperfusion injury (I/RI) may occur—a complex pathophysiological process triggered by the sudden restoration of blood flow after a period of ischemia. This damage is frequently worsened by extended cold storage (CS) (exceeding 10 h) or, in the context of donation after circulatory death (DCD), by the warm ischemia (WI) interval, during which the graft is non-perfused at body or ambient temperature prior to retrieval [5].

Moreover, the immune system, essential for survival, can turn detrimental through rejection, immunological tolerance complications, and premature graft failure [6,7]. These challenges highlight an urgent need for innovative strategies to enhance graft survival and overall transplant success.

One promising avenue of research explores harnessing endogenous cellular mechanisms to enhance transplant outcomes. Although various cellular pathways have been explored, autophagy—the mechanism through which cells recycle damaged components—has garnered much attention in recent years for its ability to influence cell survival, immune responses, and metabolic adaptation. Modulating autophagy prior to, during, and after transplantation, has shown potential for improving graft viability, reducing inflammation, and alleviating I/RI [8,9,10,11,12].

Autophagy is triggered under stressful conditions and regulates numerous cellular signaling pathways. It appears in several forms, each serving specific roles according to the cell type and physiological situation. While autophagy can protect cells from stress-induced damage, excessive or dysregulated autophagy can contribute to cell death and disease progression. This dual nature highlights its critical importance in the context of transplantation, irrespective of the graft type [2,13]. In kidney transplantation, autophagy preserves renal homeostasis and alleviates I/RI; nevertheless, its dysregulation may lead to acute kidney injury and chronic kidney disease, hence impacting graft survival [14]. Similarly, in heart transplantation, autophagy protects cardiomyocytes by clearing damaged organelles during I/RI, yet excessive activation may lead to cell death and compromise graft function [15]. In lung transplantation, studies in rat models indicate that autophagy plays a role in cold I/RI, with modulation of autophagy levels influencing graft viability [16]. These organ-specific effects highlight the necessity of tailored therapeutic strategies to optimize autophagy regulation and improve transplant outcomes.

The liver is one of the most dynamic organs in humans, and autophagy is essential for its physiology and pathology [17]. Over the past decade, extensive research has explored the role of autophagy in liver injury and disease [2,7,18]. Autophagy impairment is associated with multiple liver diseases, including alcohol-associated liver disease (AALD) [19], MASH [20], cholestasis, viral hepatitis [21,22], and hepatocellular carcinoma (HCC) [23].

Mammalian cells have reported various forms of autophagy, with macroautophagy, microautophagy, and chaperone-mediated autophagy (CMA) being the most well-characterized (Figure 1) [24]. Macroautophagy, the most studied type, facilitates the destruction of whole organelles or their impaired constituents. The process begins with the development of a membrane termed a phagophore, which enlarges to encapsulate the damaged material, resulting in a double-membrane structure referred to as an autophagosome. The autophagosome subsequently merges with a lysosome, an organelle containing digestive enzymes, to create an autolysosome. Within the autolysosome, the contained material is decomposed into fundamental components, which are recycled for energy or the synthesis of new cellular structures. This process is tightly regulated by signaling pathways such as the AMP-activated protein kinase (AMPK)-mammalian target of rapamycin (mTOR) pathway and is essential for the removal of damaged cellular components, disease prevention, and adapting to metabolic stress [25].

Another common type of autophagy, called CMA, targets soluble proteins with the KFERQ motif in particular. This selective mechanism involves the binding of these proteins to heat shock protein family A member 8 (HSPA8) and uniquely does not involve vesicle formation. Biodegradable substrates are transported into the lysosomal lumen through a spliced variant of lysosomal-associated membrane protein 2 (LAMP2A). CMA is significantly activated in response to several stressors, including DNA damage, hypoxia, and oxidative stress. Also, it is essential for preserving hepatic homeostasis through the regulation of liver metabolism and the suppression of tumorigenesis [26].

Microautophagy, which is facilitated by acidic organelles such as late endosomes, is the least comprehended type of autophagy. The process initiates as a response to the lack of amino acids and involves the degradation of minor cellular components through the formation of a membrane-bound structure within the lysosome. This process is helped by the reorganization of the lysosomal membrane and the formation of arm-like overhangs [24,25].

Specific terminology has been developed to identify the autophagic mechanisms that selectively target particular organelles, such as mitophagy for mitochondria, reticulophagy for the endoplasmic reticulum, pexophagy for peroxisomes, ribophagy for ribosomes, lipophagy for lipids, and ferritinophagy for iron-based compounds [13].

Mitophagy is essential for the removal of aged mitochondria, which normally have a lifespan of 10 to 25 days in a healthy liver [27]. This process functions as a crucial mechanism for specifically eliminating impaired mitochondria or proteins with aberrant cleavages, thus minimizing the buildup of reactive oxygen species (ROS) and altered mitochondrial DNA. In hepatic I/RI, mitophagy is regulated by both phosphatidylinositol-3-kinase (PI3K)-dependent and PI3K-independent pathways, and its impairment can lead to oxidative stress, metabolic dysfunction, and hepatocyte death [26].

Given the importance of autophagy in liver health and disease, each of these pathways represents a potential therapeutic target for improving liver transplant outcomes. This review explores the impact of autophagy modulation in liver transplantation, emphasizing its impact on graft survival, immunological regulation, and I/RI, while pinpointing significant knowledge gaps and prospective research avenues (Figure 2).

## 2. Optimizing Autophagy Regulation Prior to Liver Transplantation

A significant challenge in liver transplantation is organ preservation. During this interval, the donor liver is susceptible to several types of damage, chiefly attributable to I/RI. These injuries may severely impact graft viability and functionality following transplantation. Present preservation strategies, although somewhat effective, continue to pose difficulties in reducing cellular damage and enhancing graft quality [28].

Existing preservation techniques encompass CS, hypothermic machine perfusion (HMP), normothermic ex vivo perfusion and hypothermic oxygenated perfusion (HOPE) [29]. Meanwhile, ischemia preconditioning has garnered interest as a potential preventive approach. Preconditioning augments hepatocyte resistance to I/RI by activating cytoprotective mechanisms, including lysosome exocytosis, which mitigates intracellular acidosis, stabilizes pH, and reduces oxidative stress during ischemia [30]. This lysosomal fusion process, mediated by Ca^2^^+^-dependent signaling, is closely linked to autophagy and plays a key role in preventing cell death and improving graft quality.

Autophagy is essential in alleviating damage linked to various preservation techniques. Autophagy preserves cellular homeostasis and energy production by eliminating damaged organelles, protein aggregates, and other cellular waste products. I/RI induces characteristic cellular disturbances such as cytoplasmic acidification, sodium overload, and elevated oxidative stress [31,32], all of which impair cellular energy balance and viability. In this context, autophagy serves as a compensatory survival mechanism that restores metabolic equilibrium and prevents further injury. This is especially critical during the ischemia phase when the liver lacks oxygen and nutrients. Autophagy further regulates inflammation and apoptosis, therefore offering extra protection to the liver against damage [33].

Autophagy significantly enhances organ preservation in liver transplantation by modulating critical variables such as Sirtuin 1 (SIRT1) and High Mobility Group Box 1 (HMGB1), which are implicated in liver I/RI [34]. The regulation of these proteins is crucial for alleviating cold I/RI [3], as demonstrated by tests employing preservation solutions such as Institut Georges Lopez-1 (IGL-1), particularly when enhanced with Trimetazidine (TMZ), an autophagy inducer [4]. TMZ has been demonstrated to increase SIRT1 levels while reducing HMGB1 levels, similar to the protective effects observed during fasting. It also inhibits HMGB1-Toll-like receptor 4 (TLR4) signaling pathway, reducing inflammatory responses and improving liver graft resilience to ischemic stress.

HOPE has demonstrated the ability to trigger autophagy in liver transplants. The increased autophagy enhances the protective effects of HOPE, facilitating cellular repair and mitigating reperfusion damage. Inhibiting autophagy with chemicals such as 3-methyladenine (3-MA) reduces the advantages of HOPE, underscoring the essential function of autophagy in this preservation technique. 3-MA blocks autophagy by inhibiting the function of PI3K, a crucial regulator of autophagosome formation [4].

Autophagy aids in cellular self-repair and the recycling of damaged components, hence safeguarding liver function during storage and reducing reperfusion injury. Consequently, treatments that augment autophagy during preservation, such as oxygenated perfusion approaches like HOPE, present a promising method to boost liver graft quality and transplant success [4].

Table 1 summarizes significant research investigating autophagy regulation as a method for organ preservation before liver transplantation. These findings highlight the potential of autophagy modification as a therapeutic approach to increase liver graft quality and improve outcomes in liver transplantation.

## 3. Regulating Autophagy Dynamics for Enhanced Protection Post Liver Transplantation

### 3.1. The Dual Role of Autophagy in I/RI

Hepatocytes, the primary functioning cells of the liver, are rich in mitochondria, which are crucial for their metabolic activities [35]. During I/RI, which usually occurs throughout liver transplantation, hepatocytes are subjected to temporary interruption of blood flow, followed by restoration, leading to tissue damage [36]. Currently, I/RI closely links to the poor prognosis of large-scale liver surgeries, such as liver transplantation and hepatectomy [37]. Apoptosis and necrosis are the primary modes of hepatocyte death in this context and are exacerbated by disturbances in cellular processes such as autophagy [4]. Additionally, I/RI triggers innate immune responses post-transplantation, further complicating recovery [7].

Autophagy plays a complex, dual role in I/RI depending on the duration of ischemia. Transient ischemia activates autophagy as a protective mechanism to maintain cellular homeostasis. Autophagic flux aids in cellular recovery upon reperfusion, thereby reducing liver injury. However, in cases of prolonged ischemia, autophagy becomes inhibited, leading to exacerbated tissue damage and increased cell death [35]. The fundamental mechanism entails ischemia-induced endoplasmic reticulum (ER) stress, which has varying impacts on autophagy: transient ischemia enhances autophagy via activation of the unfolded protein response (UPR), whereas prolonged ischemia triggers a detrimental ER stress response that inhibits autophagy by downregulating essential autophagy-related proteins, such as ATG3 and ATG5 [35]. This paradox means that therapies that target either ER stress or autophagy should be used with care, since therapies that block ER stress without a specific target may stop protective autophagy during temporary ischemia while restoring beneficial autophagy during chronic ischemia. Consequently, the equilibrium between ER stress and autophagy is a pivotal factor influencing liver I/RI outcomes.

### 3.2. Therapeutic Modulation of Autophagy

Pharmacological strategies targeting autophagy, due to its intricate involvement in I/RI, have shown potential in enhancing liver transplant outcomes (Figure 3). Wang et al. investigated the use of Suberoylanilide hydroxamic acid (SAHA), an anti-tumor compound that enhances autophagy and alters immune responses in Kupffer cells, the liver’s resident macrophages. These medications aim to reduce inflammation and improve liver transplantation results by protecting against I/RI-related issues [38].

Liu et al. investigated the influence of spermidine on autophagy regulation via distinct cellular pathways, including AMPK-mTOR-ULK1 [39]. In I/RI models, spermidine improves hepatocyte survival and decreases tissue damage, highlighting its potential as a liver transplant therapy.

Moreover, the inhibition of glycogen synthase kinase 3 beta (GSK3β) and the administration of 2-Cyano-3, 12-Dioxooleana-1, and 9-Dien-28-Imidazolide (CDDO-Im) have demonstrated the capacity to activate autophagy in hepatocytes, therefore supporting the protective role of autophagy in liver I/RI. GSK3β inhibition stimulates autophagy through the activation of the AMPK/mTOR pathway, thereby enhancing hepatocyte survival and diminishing apoptosis. Concurrently, CDDO-Im activates nuclear factor erythroid 2-related factor 2 (Nrf2), resulting in an elevated expression of heme oxygenase-1 (HO-1), which further promotes autophagy and enhances mitochondrial function. Collectively, these strategies underscore the essential function of autophagy in enhancing hepatocyte viability and mitigating inflammation, therefore identifying both GSK3β inhibition and CDDO-Im as viable therapeutic targets for optimizing liver transplantation outcomes [40,41].

Pituitary adenylate cyclase-activating polypeptide (PACAP) has shown potential in enhancing liver transplant survival, especially under conditions that simulate extended CS [42]. PACAP attains these benefits via promoting autophagy, which safeguards liver cells against I/RI and enhances hepatocellular regeneration through autophagy-dependent processes. The involvement of the CREB-KLF4 pathway in PACAP-induced regulation of liver autophagy underscores its importance in re-establishing cellular homeostasis following I/RI. In warm hepatic I/RI models, PACAP demonstrated the capacity to mitigate macrophage-mediated inflammation and prevent hepatocyte death by modulating essential signaling pathways. Moreover, PACAP enhances autophagy in primary liver cells exposed to oxidative stress, protecting them from oxidative damage. The results highlight PACAP as a viable treatment strategy for mitigating transplantation-related liver damage, given its ability to augment autophagy and preserve liver cell functionality, therefore significantly enhancing transplantation outcomes [42].

### 3.3. Autophagy and Immune Responses

Autophagy is increasingly recognized as a key player in immune responses following transplantation [43]. In adaptive immunity, it participates in antigen presentation and regulates the activation, proliferation, and differentiation of immune cells [44]. Moreover, autophagy modulates innate immune signaling pathways through the degradation of inflammatory proteins, oxidative stress intermediates, and interactions with pattern recognition receptors (PRRs) [45]. Considering the significant impact of innate immunity on graft rejection, understanding the role of autophagy in this context is essential for enhancing transplant results.

### 3.4. Modulating Autophagy in Kupffer Cells and T Cells

CD8+ T cells are the primary lymphocytes responsible for graft infiltration, and their prevalence is directly associated with the intensity of acute rejection after liver transplantation [12]. Inhibiting autophagy in these T cells has been recognized as a viable strategy to reduce their survival and function, presenting a possible method to extend both graft and recipient survival.

Kupffer cells play a fundamental role in coordinating the innate immune response during I/RI. Enhancing autophagy in these cells has been shown to significantly inhibit the activation of NOD-like receptor family pyrin domain containing 3 (NLRP3) inflammasomes, key drivers of inflammation [37]. Conversely, suppression of the protein Eva-1 homologous gene A (Eva1a), an autophagy inducer, leads to significant liver damage by disrupting autophagy and exacerbating the inflammatory response, characterized by elevated levels of pro-inflammatory cytokines such as tumor necrosis factor-α (TNF-α) and interleukin-1 beta (IL-1β). Importantly, Eva1a’s regulation of autophagy is independent of the Beclin1-Vps34 pathway, which is involved in many other crucial cellular activities, underscoring its potential as a therapeutic target for mitigating inflammatory damage in liver I/RI [37].

### 3.5. Fine-Tuning Autophagy: Balancing Activation and Inhibition

While autophagy activation is generally protective in I/RI, there are instances where its downregulation may be beneficial. Bergenin, a natural compound found in several medicinal plants with antioxidant, anti-inflammatory, and hepatoprotective properties, has exhibited considerable hepatoprotective benefits in I/RI, principally related to its antioxidant capabilities and capacity to alter critical molecular pathways [8]. Bergenin’s primary mechanism involves the removal of ROS produced during I/RI and subsequently suppresses the activation of the P38 mitogen-activated protein kinase (P38 MAPK) pathway, a key component in inflammation and cell death [8]. Bergenin reduces excessive autophagy by enhancing B-cell lymphoma 2 (Bcl-2) expression, which interacts with Beclin-1 and hinders the conversion of LC3-I to LC3-II. This reduction in autophagy, along with the suppression of apoptosis and necrosis, enhances Bergenin’s protective effects on hepatocytes.

Hepatic I/RI involves multiple pathways, including ROS/MAPK, ROS/c-Jun N-terminal kinase (JNK)/Bcl-2, and HMGB1/TLR4/nuclear factor-kappa B (NF-kappaB). Among these, PI3K/Akt pathway, a key regulator of cell survival and apoptosis, is critical for protecting against I/RI [46]. Shikonin, an anti-inflammatory and anticancer compound, protects liver tissues during I/RI by activating the PI3K/Akt pathway. This activation enhances expression of the anti-apoptotic protein Bcl-2 while decreasing pro-apoptotic proteins like Bcl-2-associated X protein (Bax), caspase 9, and caspase 3, thus reducing apoptosis. Furthermore, shikonin facilitates the assembly of the Bcl-2/Beclin-1 complex, resulting in decreased Beclin-1 expression and the suppression of autophagy [8].

### 3.6. Mitophagy in Liver Transplantation

Alongside general autophagy, mitophagy, a selective form of autophagy that targets mitochondria, also plays a critical role in I/RI. PINK1/Parkin-mediated mitophagy has been shown to mitigate I/RI by reducing mitochondrial ROS and preventing NLRP3 inflammasome activation [47]. In Kupffer cells, this pathway helps regulate mitochondrial quality, thereby suppressing excessive inflammation and apoptosis during I/R events. 25-Hydroxycholesterol also decreases NLRP3 inflammasome activation and mitigates hepatic I/RI by enhancing mitophagy, although additional investigation is required to comprehensively elucidate this association [48].

Polyethylene glycol 35 kDa (PEG35) has shown potential in experimental settings as a protective mediator for mitochondria, significantly mitigating liver damage. In hypoxia/reoxygenation experiments with HepG2 cells, PEG35 preconditioning reduced excessive ROS production, mitigated ATP depletion, and stabilized mitochondrial membrane potential. Additionally, PEG35 promotes autophagy-related proteins and genes associated with mitochondrial biogenesis and fusion, underscoring its role in improving mitochondrial quality control [49].

Numerous autophagy modulators demonstrate diverse impacts on graft survival and associated I/RI damage, as outlined in Table 2.

## 4. Autophagy and Xenotransplantation

Xenotransplantation, especially pig-to-human transplants, offers a viable solution to the worldwide organ donor deficit. This approach is impeded by immunological obstacles, with rejections occurring in three main forms: hyperacute rejection (HAR), acute humoral xenograft rejection (AHXR), and acute cellular rejection. HAR, occurring within minutes to hours post-transplant, is induced by natural antibodies, whereas AHXR and cellular rejection, developing over days or weeks, are mediated by T and B lymphocytes, natural killer (NK) cells, and macrophages. Mitigating these immune responses poses a significant challenge to the efficacy of xenotransplantation [6].

In xenotransplantation, autophagy is essential for modulating immunological rejection, indicating novel therapeutic approaches to enhance xenograft viability and transplantation results. Proteins associated with autophagy, including ATG16L1-NOD1, ATG5/12-RIG-I, and TRAF6-Beclin-1, directly engage with PRRs and their downstream signaling components, indicating a possible role for autophagy in modulating innate immune responses after organ donation [57]. Specifically, autophagy can modulate the inflammatory response by degrading pro-inflammatory cytokines and oxidative intermediates or by targeting NLRP3, type II interferon, and PI3K signaling after graft transplantation. These mechanisms are especially important in xenotransplantation, where inflammation and oxidative damage significantly contribute to rejection. Nonetheless, autophagy may also accelerate the inflammatory response, oxidative damage, and cellular death under extreme stress settings, such as prolonged ischemia or substantial reperfusion injury. The adverse effects impede the enhancement of xenotransplant results, particularly when faced with organ stressors such as mechanical injury, ischemia, and hypoxia, frequently seen in xenograft models.

Although the modification of immune responses by autophagy is well recognized, its specific mechanisms within the context of xenotransplantation are not well defined. To comprehend how autophagy affects immunological rejection, particularly in organ transplantation, it is crucial to identify suitable animal models that replicate the many stresses associated with transplantation. In this context, progress in high-throughput sequencing and organoid research may elucidate the interactions among autophagy, innate immunity, and rejection mechanisms in xenotransplantation [6,57].

## 5. Conclusions

Autophagy, a cellular degradation mechanism that eliminates harmful intracellular components, has a multifaceted and context-sensitive function in liver transplantation, contributing to protection against preservation injury, modulation of immune responses, and mitigation of I/RI [58]. This review has outlined key regulatory mechanisms and ramifications of autophagy in the context of liver transplantation.

Despite encouraging progress in the field, several challenges remain. Numerous studies concentrate on discrete phases of the transplantation process, overlooking the dynamic and stage-specific roles of autophagy during preservation, reperfusion, and recovery. Furthermore, investigations have primarily focused on macroautophagy, whereas other subtypes, such as mitophagy and CMA remain comparatively underexamined. A comprehensive knowledge of these specific autophagic mechanisms, especially concerning time and physiological context, is crucial for developing effective interventions.

While autophagy regulation demonstrates therapeutic potential, its clinical use necessitates thorough assessment of drug delivery methods, discovery of reliable biomarkers, and consideration of individual patient factors. Alongside experimental compounds, other authorized pharmacological and nutraceutical treatments have demonstrated the ability to stimulate protective autophagy. Therapeutics like metformin [59] and adenosine A2A receptor [60] modulators augment autophagic flux and provide cytoprotection in models of liver damage. Nutraceuticals such as resveratrol have exhibited the capacity to activate autophagy and mitigate oxidative stress, while also enhancing hepatocyte survival after I/RI via regulating the unfolded protein response [61]. These findings endorse the feasibility of repurposing established medicines to expedite the practical use of autophagy-based approaches in liver transplantation.

Progressing in this domain necessitates a more cohesive and comprehensive research framework which needs the diverse forms of autophagy, their systemic effects, and the ideal timing of intervention along the transplantation timeline.

## Figures and Tables

**Figure 1 biomolecules-15-00797-f001:**
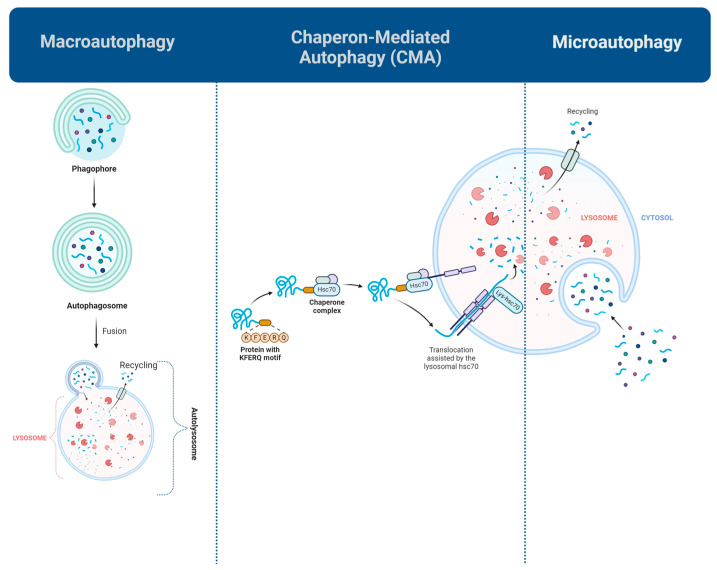
Main types of autophagy, Macroautophagy, is a crucial cellular process that enables the degradation and recycling of old or damaged components. During this process, cytoplasmic material targeted for degradation is enclosed within a double-membrane structure, forming an autophagosome. The autophagosome then fuses with a lysosome, an organelle containing hydrolytic enzymes. These enzymes break down the enclosed contents into basic molecular components, which the cell can then recycle for energy production or the synthesis of new cellular structures. Chaperone-mediated autophagy (CMA) acts selectively on target proteins by the KFERQ domain, which is recognized by heat shock cognate 71 kDa protein (Hsc70) and then recruited to the lysosome by lysosomal-associated membrane protein type 2 (LAMP2A). Microautophagy is the simplest form of autophagy in which the target cargo is directly engulfed by lysosomes (Created in Biorender. Kolahdouzmohammadi, M. (2025) https://BioRender.com/6pbz6k3 (accessed on 28 May 2025)).

**Figure 2 biomolecules-15-00797-f002:**
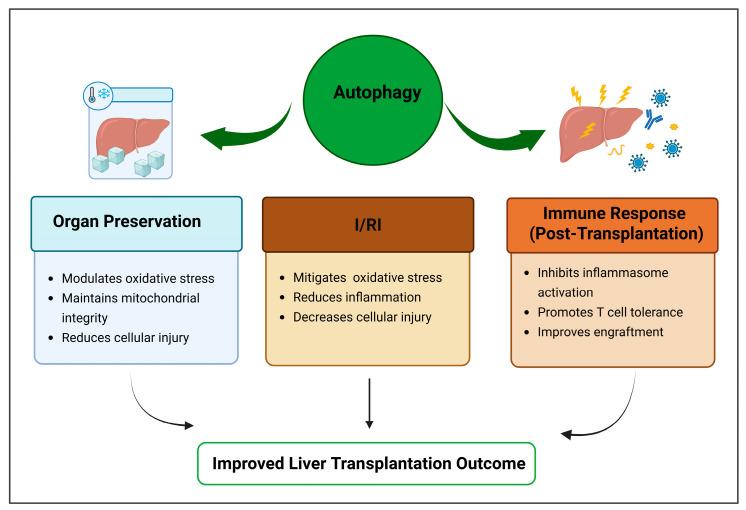
Role of autophagy in key stages of liver transplantation (Created in Biorender. Kolahdouzmohammadi, M. (2025) https://BioRender.com/3ahmz02 (accessed on 28 May 2025)).

**Figure 3 biomolecules-15-00797-f003:**
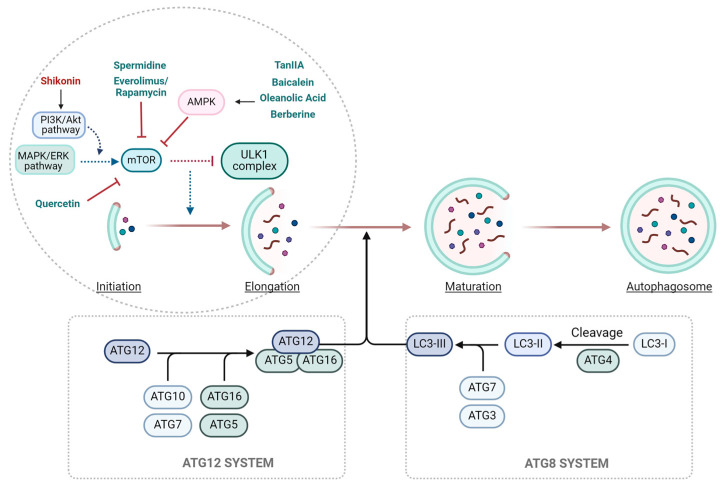
Macroautophagy as a main liver transplantation target. This pathway operates through functional complexes. The Atg1/ULK complex responds to upstream signals and mTORC1 levels, as an inhibitor of the mammalian ULK1 complex directly affects this complex. The second complex is class III phosphatidylinositol 3-kinase (PI3K), which mediates vesicle nucleation. The elongation and maturation of the phagophore rely on conjugation cascade complexes, with Light Chain 3 (LC3) and various *Atg* genes serving as critical components. Common autophagy modulators used in optimizing liver transplantation, highlighted within a dashed circle, generally act during the early stages of this pathway. T-bar arrows indicate inhibition (Created in Biorender. Kolahdouzmohammadi, M. (2025) https://BioRender.com/6pbz6k3 (accessed on 28 May 2025)).

**Table 1 biomolecules-15-00797-t001:** Impact of autophagy modulation on liver preservation.

Study (Author, Year)	Experimental Model	Groups & Sample Size	Preservation Mode	Intervention (Autophagy Modulator)	Modulator Role (Inducer/Inhibitor)	Main Results (Quantitative)
Zaouali et al., 2017 [3]	Isolated perfused rat liver model (Steatotic and Non-Steatotic)	Three Groups (16 Zücker rats per group, 8 Obese and 8 Lean): Control, IGL-1, IGL-1 + TMZ	CS	TMZ (10^−6^ M) in IGL-1 solution	Inducer. (Induces autophagy via SIRT1)	IGL-1 + TMZ showed significant reductions in liver injury markers (AST, ALT, and GLDH) in Non-Steatotic and Steatotic Livers.
Zeng et al., 2019 [4]	Mouse (C57BL/6)	Eight Groups (six mice per group): Control, CS, HNPE, HOPE, CS-Reperfusion, HNPE-Reperfusion, HOPE-Reperfusion, 3-MA	CS	3-MA (30 mg/kg 1 h before WI)	Inhibitor. (Attenuates protective effect of HOPE)	HOPE showed significant reduction in liver injury markers (AST, ALT) and increased in autophagy markers (LC3B-II, Atg5, ULK1) vs. CS and HNPE.
Nakamura et al., 2017 [34]	Mouse (C57BL/6)	Three Groups (4–6 mice per group): Control, AdHO-1, Adβ-gal	CS	HO-1 overexpression (via genetically modified macrophages)	Inducer. (Stimulates SIRT1)	AdHO-1-treated OLT in mouse and human models showed reduced liver injury marker (ALT) and enhanced autophagy markers (SIRT1 and LC3B) compared to controls with low HO-1 expression.
	Human (OLT model)	Two Groups: Low-HO-1, High-HO-1 (51 OLT patients)		EX527 (Administered 30 min before OLT surgery)	Inhibitor. (Blocks SIRT1, reduces autophagy, diminishes HO-1-mediated hepatoprotection)	

AdHO-1: Ad encoding HO-1, HNPE: Hypothermic deoxygenated (nitrogenated) perfusion, OLT: Orthotopic liver transplantation, and ULK1: Unc-51 Like Autophagy Activating Kinase 1.

**Table 2 biomolecules-15-00797-t002:** Autophagy modulation and I/R injury.

Study (Author, Year)	Experimental Model	Groups & Sample Size	I/RI Induction Method	Intervention (Autophagy Modulator)	Type of Autophagy	Modulator Role (Inducer/Inhibitor)	Main Results (Quantitative)
Lee et al., 2016 [50]	In vitro: (AML12 cells)	Three Groups: Control, IRI, and IRI + Everolimus	Following one hour in Krebs–Henseleit buffer containing antimycin A and 2-deoxyglucose, the cells were reperfused with complete media.	Everolimus (100 nM)	Macroautophagy	Inducer. (promotes autophagy by inhibiting mTOR)	IRI + Everolimus showed enhanced autophagy markers, and decreased apoptosis compared to IRI alone.
	In vivo: (BALB/c mice)	Four Groups (15 mice per group): Sham + Saline, Sham + Everolimus, IRI + Saline, and IRI + Everolimus	45 min of hepatic ischemia followed by reperfusion.	Everolimus (1 mg/kg, 24 h before and immediately after IRI induction)			IRI + Everolimus showed significant IL-6 and TNF-α reduction, lower liver enzyme levels (AST, ALT, and ammonia), and improved liver histopathology compared to IRI + Saline.
Wang et al., 2020 [38]	In vitro: (Kupffer Cells)	Six Groups: Normal, lipopolysaccharide (LPS), LPS + DMSO, LPS + SAHA, LPS + Scramble-siRNA, and LPS + SAHA + ATG5-siRNA	LPS stimulation to induce inflammation.	SAHA (3 μM)	Macroautophagy	Inducer. (promotes autophagy through inhibition of the AKT/mTOR pathway)	LPS + SAHA inhibited M1 polarization of Kupffer cells and showed increased autophagy markers compared to LPS alone.
	In vivo: (Sprague-Dawley rats)	Eight Groups (five rats per group): Sham, IRI, IRI + DMSO, IRI + SAHA, IRI + SAHA + Clodronate liposomes, IRI + SAHA + Chloroquine, IRI + Scramble-shRNA, and IRI + SAHA + ATG5-shRNA	Modified Kamada’s two-cuff technique.	SAHA (50 mg/kg, 12 h prior to OLT and continued q12 h after reperfusion)			IRI + SAHA showed decreased levels of IL-1β, TNF-α, AST, and ALT, along with enhanced histopathology, and reduced M1 polarization of Kupffer cells compared to IRI alone.
Liu et al., 2019 [39]	In vivo: (C57BL/6 mice)	Four Groups (six mice per group): Sham, Sham + Spermidine, IRI, and IRI + Spermidine	60 min of partial hepatic ischemia (70%) followed by reperfusion.	Spermidine (3 mM, 2 mL/day for four weeks before IRI induction)	Macroautophagy	Inducer. (stimulates autophagy through the AMPK-mTOR-ULK1 pathway)	IRI + Spermidine showed reduced IL-6, TNF-α, AST, ALT, and improved histopathology compared to IRI alone.
Lin et al., 2017 [51]	In vivo: (Wistar rats)	Four Groups (six rats per group): Sham, OLT, OLT + Berberine, and OLT + Berberine + EX527	OLT using the Kamada’s two-cuff technique followed by reperfusion.	Berberine (100 mg/kg/day, for two weeks before OLT)	Sirt1/FoxO3α Induced Autophagy	Inducer. (activates autophagy via the Sirt1/FoxO3α pathway)	OLT + Berberine showed reduced ALT, AST, improved histopathology, and decreased oxidative stress compared to OLT alone.
Jiang et al., 2019 [52]	In vivo: (C57BL/6 mice)	Six Groups (six mice per group): Sham, IRI, Ischemic preconditioning (IPC), Rapamycin, IPC + Rapamycin, and IPC + Rapamycin + 3-MA	90 min of segmental (70%) hepatic ischemia followed by reperfusion.	Rapamycin (1 mg/kg, 1 h before ischemia)	Macroautophagy	Inducer. (promotes autophagy by inhibiting mTOR)	IPC + Rapamycin showed reduced ALT, and improved histopathology, compared to IPC alone.
Nakamura et al., 2017 [34]	In vivo: (C57BL/6 mice)	Three Groups (six mice per group): BMMs, BMM (ad encoding HO-1), and BMM (Adβ-gal)	20 h of CS in UW solution followed by OLT and reperfusion.	HO-1 (2.5 × 10^9^ pfu) via adoptive transfer of Ad encoding HO-1-transfected BMMs	Macroautophagy	Inducer. (activates autophagy through the SIRT1)	BMM (Ad encoding HO-1) showed reduced ALT levels, and improved histopathology (lower Suzuki score) compared to BMM (Adβ-gal)
Kong et al., 2019 [40]	In vitro: (primary hepatocytes)	Three Groups: H/R + DMSO, H/R + SB216763, and H/R + SB216763 + 3-MA	Cells underwent 4 h of hypoxia (1% O_2_) followed by 2 h of reoxygenation.	SB216763	Macroautophagy	Inducer. (activates autophagy through the inhibition of GSK3β)	H/R + SB216763 showed reduced LDH release (lower cell death), and increased autophagy compared to control group.
	In vivo: (C57BL/6 mice)	Four Groups (4–6 mice per group): Sham, DMSO + IRI, IRI + SB216763, and IRI + SB216763 + 3-MA	90 min of partial hepatic ischemia (70%) followed by reperfusion.	SB216763 (20 mg/kg, 1 h before ischemia)			IRI + SB216763 showed reduced AST, ALT, improved histopathology, compared to DMSO + IRI.
Xu et al., 2017 [41]	In vitro: (primary hepatocytes)	Two Groups: DMSO + H/R and CDDO-Im + H/R	Cells underwent 4 h of hypoxia (1% O_2_) followed by 2 h of reoxygenation.	CDDO-Im (200 μM)	Macroautophagy	Inducer. (activates autophagy through the Nrf2/HO-1 pathway)	CDDO-Im + H/R showed reduced LDH release (less cell death), and enhanced mitochondrial function (lower mtDNA release, improved ROS clearance) compared to DMSO + H/R.
	In vivo: (C57BL/6 mice)	Three Groups (4–6 mice per group): Sham, IRI+DMSO, and IRI + CDDO-Im.	90 min of partial hepatic ischemia followed by reperfusion.	CDDO-Im (2 mg/kg, 3 h before ischemia)			IRI + CDDO-Im. showed reduced AST, ALT, and improved histopathology, compared to IRI + DMSO.
Xue et al., 2020 [42]	In vitro: (primary hepatocytes)	Four Groups: PBS, H_2_O_2_, H_2_O_2_ + PACAP+DMSO, and H_2_O_2_ + PACAP + 3-MA	Hydrogen peroxide (H_2_O_2_) 0.4 mM.	PACAP38 (10 nM)	Macroautophagy	Inducer. (activates autophagy through the CREB-KLF4 pathway)	H_2_O_2_ + PACAP showed reduced LDH release
	In vivo: (C57BL/6 mice)	Four Groups (12 mice per group): Sham, PBS, PACAP + DMSO, and PACAP + 3-MA.	20 h of storage at 4 °C in UW solution followed by syngeneic OLT and reperfusion.	PACAP38 (1 mg/kg at the time of liver graft procurement and before reperfusion)			PACAP + DMSO showed reduced ALT, improved histopathology, and decreased necrosis compared to PBS group.
Wu et al., 2017 [11]	In vitro: (primary hepatocytes)	Seven Groups: Control, QE, DMSO, H/R, H/R + QE 5 μM, H/R + QE 10 μM, and H/R + QE 20 μM	Cells underwent 24 h of hypoxia (3% O_2_) followed by 2 h of reoxygenation.	Quercetin (QE) (20 μM)	Macroautophagy	Inhibitor. (inhibits autophagy through ERK/NF-κB pathway)	H/R + QE showed decreased apoptosis, reduced autophagy markers, and enhanced cell viability compared to H/R alone.
	In vivo: (Balb/c mice)	Eight Groups (5 and 24 mice per group): Control, Vehicle, Low-QE, High-QE, Sham, IRI, QE100 + IRI, and QE200 + IRI	45 min of 70% hepatic WI followed by reperfusion.	Quercetin (100 mg/kg or 200 mg/kg, before IRI)			IRI + QE showed reduced ALT, AST, improved histopathology, reduced pro-inflammatory cytokines (TNF-α, IL-6), and decreased apoptosis compared to IRI alone.
Wang et al., 2019 [9]	In vivo: (Balb/c mice)	Five Groups (18 mice per group): Sham, CMC, IRI, Low-OA (30 mg/kg), and High-OA (60 mg/kg)	60 min of partial (70%) hepatic ischemia followed by reperfusion.	Oleanolic Acid (OA) (30 mg/kg or 60 mg/kg, for seven days before surgery)	Macroautophagy	Inhibitor. (inhibits autophagy through JNK phosphorylation and HMGB1 suppression)	Low-OA and High-OA groups showed reduced ALT, AST, and improved histopathology, compared to IRI group.
Liu et al., 2016 [53]	In vitro: (primary rat hepatocytes)	Four Groups: Control, H/R, H/R + Baicalein, and H/R + Baicalein + 3-MA	Cells underwent 6 h of hypoxia (1% O_2_) followed by 2 h of reoxygenation.	Baicalein (5 μM)	Macroautophagy	Inducer. (activates autophagy via HO-1)	H/R + Baicalein showed improved cell viability, and reduced LDH release compared to H/R alone.
	In vivo: (Sprague-Dawley rats)	Three Groups (six rats per group): Sham, IRI, and Baicalein + IRI	60 min of partial hepatic WI followed by reperfusion.	Baicalein (100 mg/kg, 1 h before ischemia)			Baicalein + IRI showed reduced ALT, AST, improved histopathology, and reduced necrosis, compared to IRI alone.
Qin et al., 2016 [54]	In Vivo: (C57BL/6 mice)	Two Main Groups: Sham (control, one day-STS, two day-STS, and three day-STS), IRI (control, one day-STS, two day-STS, and three day-STS).	90 min of partial hepatic ischemia followed by reperfusion.	Short-term starvation (STS) for one, two, or three days before IRI	Macroautophagy	Inducer. (enhances autophagy via Sirt1 activation)	IRI + STS showed reduced ALT, AST, improved histopathology, decreased IL-1β, TNF-α, and apoptosis markers compared to IRI alone.
Xiang et al., 2020 [8]	In vitro: (LO2 hepatocytes)	Four Groups: Control, H/R, H/R + Bergenin, and H/R + Bergenin + GW9662	Cells underwent 24 h of hypoxia (3% O_2_) followed by 2 h of reoxygenation.	Bergenin (3 mM)	Peroxisome proliferators activated receptor γ (PPAR-γ) pathway mediated autophagy	Inhibitor. (inhibits autophagy through the PPAR-γ Pathway)	H/R + Bergenin showed enhanced cell viability, reduced apoptosis, and decreased autophagy markers compared to H/R.
	In vivo: (Balb/c mice)	Five Groups (24 mice per group): Sham, IRI, IRI + Bergenin (10 mg/kg), IRI + Bergenin (20 mg/kg), and IRI + Bergenin (40 mg/kg).	60 min of partial hepatic ischemia (70%) followed by reperfusion.	Bergenin (10, 20, 40 mg/kg, for three days) (once before the operation, once per day)			IRI + Bergenin showed reduced ALT, AST, improved histopathology, and decreased inflammation, compared to IRI alone.
Chen et al., 2019 [12]	In vivo: (Lewis and Brown Norway rats)	Three Groups (five rats per group): Syngeneic control, Allogeneic, and Allogeneic + 3-MA	OLT using the Two-Cuff Method.	3-MA (24 mg/kg, every three days, starting one day before transplantation, dosage reduced by half every nine days)	Macroautophagy	Inhibitor.	Allogeneic + 3-MA group showed prolonged survival time, reduced ALT, AST, and improved histopathology compared to Allogeneic group without 3-MA.
Wang et al., 2021 [37]	In vitro: (Kupffer cells)	Mixed Groups: Control, IRI + Eva1a OE, and IRI + Eva1a KD	-	Eva1a Overexpression (OE) and Knockdown (KD).	Macroautophagy	Inducer. (Eva1a OE Promotes autophagy by interacting with ATG16L1).	IRI + Eva1a KD showed increased TNF-α, increased ALT, AST, compared to IRI alone.
	In vivo: (C57BL/6 mice)	Four Groups (4–6 mice per group): Sham, IRI, IRI + Eva1a OE, and IRI + Eva1a KD.	90 min of partial hepatic ischemia (70%) followed by reperfusion.	Eva1a-OE and Knockdown (KD).			
Yu et al., 2019 [55]	In Vivo: (BALB/c mice)	Six Groups (six mice per group): Control, Sham, L-THP (40 mg/kg), IRI, L-THP (20 mg/kg) + IRI, and L-THP (40 mg/kg) + IRI	45 min of partial hepatic ischemia (70%) followed by reperfusion.	Levo-tetrahydropalmatine (L-THP) (20 mg/kg and 40 mg/kg, for five days before IRI).	Macroautophagy	Inhibitor. (suppresses autophagy via inhibition of ERK/NF-κB signaling pathway)	L-THP + IRI showed reduced AST and ALT levels, decreased TNF-α and IL-6, and improved liver architecture compared to IRI alone.
Liu et al., 2017 [46]	In vivo: (Balb/c mice)	Four Groups (18 mice per group): Control, IRI, IRI + Shikonin 7.5 mg/kg, and IRI + Shikonin 12.5 mg/kg	45 min of segmental (70%) hepatic ischemia followed by reperfusion.	Shikonin (7.5 mg/kg or 12.5 mg/kg, 2 h before ischemia).	Macroautophagy	Inhibitor. (inhibits autophagy via the PI3K/Akt pathway)	IRI + Shikonin showed reduced ALT, AST, improved histopathology, and decreased IL-1β, TNF-α, IL-6, compared to IRI alone.
Wang et al., 2018 [56]	In vivo: (C57BL/6 mice)	Four groups (six mice per group): Vehicle, TanIIA, IRI + Vehicle, and IRI + TanIIA	60 min of partial hepatic ischemia (70%) followed by reperfusion	Tanshinone IIA (TanIIA) (20 mg/kg/day, for three days before IRI).	Macroautophagy	Inducer. (activates autophagy through the MEK/ERK/mTOR pathway)	IRI + TanIIA showed reduced ALT, AST, improved histopathology, and reduced ROS production, compared to IRI+Vehicle group.
Cao et al., 2021 [48]	In vivo: (Sprague-Dawley rats)	Six Groups (six rats per group): Sham, 25HC, IRI (3) + Vehicle, IRI (3) + 25HC, IRI (24) + Vehicle, and IRI (24) + 25HC.	60 min of partial hepatic ischemia (70%) followed by reperfusion.	25-Hydroxycholesterol (25HC) (30 mg/kg 4 h before ischemia).	Mitophagy	Inducer. (activates PINK1/Parkin-dependent mitophagy)	IRI + 25HC showed reduced ALT, AST, improved histopathology, decreased NLRP3 activation, and reduced IL-1β, compared to IRI + Vehicle.
Teixeira da Silva et al., 2022 [49]	In vitro: (HepG2 cells)	Six groups: Control, Control + 1% PEG35, Control + 5% PEG35, H/R, 1% PEG35 + H/R, and 5% PEG35 + H/R	H/R: 2 h of hypoxia (hypoxia mimic solution) followed by 2 h of reoxygenation (complete medium)	PEG35 (1% or 5%, preconditioning for 1 h before H/R)	Mitophagy	Inducer. (enhances mitochondrial biogenesis and recovery of membrane potential).	5% PEG35 + H/R showed increased cell viability, preserved mitochondrial membrane potential, reduced ROS, maintained ATP levels, and improved mitochondrial dynamics by enhancing fusion and reducing fission compared to H/R alone.
Xu et al., 2020 [47]	In vitro: (Kupffer cells)	Two Groups: Control (Vector and PINK1 OE), H/R (Vector and PINK1 OE)	Cells were exposed to 4 h of hypoxia (1% O_2_) followed by 4 h of reoxygenation.		Mitophagy	Inducer. (promotes PINK1/Parkin-dependent mitophagy)	H/R + PINK1 OE showed enhanced autophagy, decreased IL-1β, and TNF-α compared to H/R alone.
	In vivo: (C57BL/6 mice)	Two Groups (4–6 mice per group): Vector (Control), PINK1 OE	60 min of partial hepatic ischemia followed by reperfusion.	PINK1-OE (50 μg pcDNA3.1-mPINK1 plasmid 72 h before IRI).			PINK1 OE showed reduced ALT, AST, and improved histopathology compared to control group.

BMM: Bone marrow–derived macrophages, CMC: Carboxymethylcellulose, DMSO: Dimethyl sulfoxide, H/R: Hypoxia/Reoxygenation, and LDH: Lactate Dehydrogenase, UW: University of Wisconsin.

## Data Availability

The original contributions presented in this study are included in the article. Further inquiries can be directed to the corresponding author.

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
