# Peer review of "Tuning Autophagy for Improved Liver Transplant Outcomes: Insights from Experimental Models"

_biomolecules, 2025, doi:10.3390/biom15060797_

Round 1

Reviewer 1 Report

Comments and Suggestions for Authors

The evidence which has been obtained from laboratory based research which forms the basis for this particular review type manuscript is very limited and hence markedly constrains any conclusions which can be drawn. Hence the authors need to address the following-

1) The manuscript title needs to more adequately reflect the type of research which has been performed to date (ie laboratory type research involving either cell lines or a range of rodent animal models- most of which did not involve liver transplantation).

2) There are statements that have been made in the Introduction section which are not strictly correct, for eg the statement between lines 33-35. The I/RI syndrome occurs after liver reperfusion and is associated with a number of factors which may contribute to its occurrence in clinical practice

3) The references which are used to make the case for modulating autophagy prior to, during or post transplantation (6,8,9), all seem to be review type articles which does not seem appropriate

4) The data that is summarized into both Tables 1 and 2 is limited and does not afford the reader the opportunity to adequately gauge the limitations of each particular research study. For example in Table 1, additional information needs to be included on the numbers of animals in each of the treatment arms (including as to whether the animals underwent liver transplantation or some other method of liver ischemia was used). The outcomes also need to be more adequately encapsulated-for example when mention is made or 'reduces IRI'-how was this measured and what was the percentage decrease compared to the control animals?

5) The same applies to Table 2. In each case it is stated that the model is I/RI. This needs to be spelt out as to what was done to induce I/RI in each particular model (because there is every chance that this varied between each of the animal studies). It also needs to be spelt out as to which of the models involved liver transplantation and precisely what type of liver surgery was being performed in the non transplant studies.

6) The Conclusions section is far too long and needs to be a lot more focused. Some of the limitations which have been mentioned in the Conclusions section need to form part of a far more comprehensive discussion on the limitations of all of the published experimental research to date (of which there are a number, most of which have not been mentioned). It is speculative to hypothesize that this can all translate into the clinical liver transplantation arena at this point in time and even more speculative to be hypothesizing about xenotransplantation (which is in its infancy and in itself considered experimental currently). 

Author Response

We truly appreciate the reviewers for their insightful and critical feedback. We value the time and effort they dedicated to evaluating our work. We have meticulously addressed all concerns highlighted and implemented the necessary edits to enhance the clarity and quality of the manuscript. We present a detailed response to each comment, with our edits and rationales explicitly articulated.

Comment1: The manuscript title needs to more adequately reflect the type of research which has been performed to date (ie laboratory type research involving either cell lines or a range of rodent animal models- most of which did not involve liver transplantation).

Response1: Based on the reviewers’ valuable comments we have changed the title to represent our study better to: “Tuning Autophagy for Improved Liver Transplant Outcomes: Insights from Experimental Models”

Comment2: There are statements that have been made in the Introduction section which are not strictly correct, for eg the statement between lines 33-35. The I/RI syndrome occurs after liver reperfusion and is associated with a number of factors which may contribute to its occurrence in clinical practice.

Response2: We have revised line 33 to 43 based on the reviewer’s valuable comment to:

“Line 34-43: Following reperfusion of the transplanted liver, ischemia/reperfusion injury (I/RI) may occur — a complex pathophysiological process triggered by the sudden restoration of blood flow after a period of ischemia. This damage is frequently worsened by extended cold storage (CS) (exceeding 10 hours) or, in the context of donation after circulatory death (DCD), by the warm ischemia (WI) interval, during which the graft is non-perfused at body or ambient temperature prior to retrieval [5].

Moreover, the immune system, essential for survival, can turn detrimental through rejection, immunological tolerance complications, and premature graft failure [6,7]. These challenges highlight an urgent need for innovative strategies to enhance graft survival and overall transplant success.”

Comment3: The references which are used to make the case for modulating autophagy prior to, during or post transplantation (6,8,9), all seem to be review type articles which does not seem appropriate

Response3: The references have been updated based on the reviewer’s valuable comment.

“Line 50: references 8-12”

Comment4: The data that is summarized into both Tables 1 and 2 is limited and does not afford the reader the opportunity to adequately gauge the limitations of each particular research study. For example in Table 1, additional information needs to be included on the numbers of animals in each of the treatment arms (including as to whether the animals underwent liver transplantation or some other method of liver ischemia was used). The outcomes also need to be more adequately encapsulated-for example when mention is made or 'reduces IRI'-how was this measured and what was the percentage decrease compared to the control animals?

Response4: We have revised the entire table and included additional information in response to the reviewer’s valuable comment. The updated format offers greater detail and provides readers with more comprehensive insights.

Comment5: The same applies to Table 2. In each case it is stated that the model is I/RI. This needs to be spelt out as to what was done to induce I/RI in each particular model (because there is every chance that this varied between each of the animal studies). It also needs to be spelt out as to which of the models involved liver transplantation and precisely what type of liver surgery was being performed in the non transplant studies.

Response5: We have revised the entire table and included additional information in response to the reviewer’s valuable comment. The updated format offers greater detail and provides readers with more comprehensive insights.

Comment6: The Conclusions section is far too long and needs to be a lot more focused. Some of the limitations which have been mentioned in the Conclusions section need to form part of a far more comprehensive discussion on the limitations of all of the published experimental research to date (of which there are a number, most of which have not been mentioned). It is speculative to hypothesize that this can all translate into the clinical liver transplantation arena at this point in time and even more speculative to be hypothesizing about xenotransplantation (which is in its infancy and in itself considered experimental currently).

Response6: We have revised the entire conclusion section to make it more concise and focused.

“Line352-377: Autophagy, a cellular degradation mechanism that eliminates harmful intracellular components, has a multifaceted and context-sensitive function in liver transplantation, contributing to protection against preservation injury, modulation of immune responses, and mitigation of I/RI [59]. This review has outlined key regulatory mechanisms and ramifications of autophagy in the context of liver transplantation. 

 Despite encouraging progress in the field, several challenges remain. Numerous studies concentrate on discrete phases of the transplantation process, overlooking the dynamic and stage-specific roles of autophagy during preservation, reperfusion, and recovery.  Furthermore, investigations have primarily focused on macroautophagy, whereas other subtypes—such as mitophagy and CMA—remain comparatively underexamined.  A comprehensive knowledge of these specific autophagic mechanisms, especially concerning time and physiological context, is crucial for developing effective interventions.

 While autophagy regulation demonstrates therapeutic potential, its clinical use necessitates thorough assessment of drug delivery methods, discovery of reliable biomarkers, and consideration of individual patient factors.  Alongside experimental compounds, other authorized pharmacological and nutraceutical treatments have demonstrated the ability to stimulate protective autophagy.  Therapeutics like metformin [60] and adenosine A2A receptor [61] modulators augment autophagic flux and provide cytoprotection in models of liver damage.  Nutraceuticals such as resveratrol have exhibited the capacity to activate autophagy and mitigate oxidative stress, while also enhancing hepatocyte survival after I/RI via regulating the unfolded protein response [62].  These findings endorse the feasibility of repurposing established medicines to expedite the practical use of autophagy-based approaches in liver transplantation.

Progressing in this domain necessitates a more cohesive and comprehensive research framework—one that considers the diverse forms of autophagy, their systemic effects, and the ideal timing of intervention along the transplantation timeline.”

Reviewer 2 Report

Comments and Suggestions for Authors

see atatched

Author Response

Reviewer 2:

We truly appreciate the reviewers for their insightful and critical feedback. We value the time and effort they dedicated to evaluating our work. We have meticulously addressed all concerns highlighted and implemented the necessary edits to enhance the clarity and quality of the manuscript. We present a detailed response to each comment, with our edits and rationales explicitly articulated.

Comment1: The authors should mention the role of pre-conditioning in organ protection from ischemia-reperfusion damage and cell death. There are plenty of works and the authors may choose the most relevant and updated. In this context, I may suggest to have a read at the discussion of the paper by Carini et al. “Preconditioning-induced cytoprotection in hepatocytes requires Ca(2+)-dependent exocytosis of lysosomes. J Cell Sci. 2004 Mar 1;117(Pt 7):1065-77. doi: 10.1242/jcs.00923” where it is highlighted the role of lysosome exocytosis in preventing cell acidification during ischemia, which is likely linked to autophagy.

Response1: We have added the requested information based on the reviewer’s valuable comment:

Line 137-142: Meanwhile, ischemia preconditioning has garnered interest as a potential preventive approach. Preconditioning augments hepatocyte resistance to I/RI by activating cytoprotective mechanisms, including lysosome exocytosis, which mitigates intracellular acidosis, stabilizes pH, and reduces oxidative stress during ischemia [30]. This lysosomal fusion process, mediated by Ca²⁺-dependent signaling, is closely linked to autophagy and plays a key role in preventing cell death and improving graft quality.”

Comment2: Linked to the above, the Authors could spend few words to illustrate what is the cell damage (e.g., acidification, oxidative stress) associated with ischemia reperfusion and how these processes impact autophagy as a stress response.

Response2: We have added the requested information based on the reviewer’s valuable comment:

“Line 145-149: I/RI induces characteristic cellular disturbances such as cytoplasmic acidification, sodium overload, and elevated oxidative stress [31,32], all of which impair cellular energy balance and viability. In this context, autophagy serves as a compensatory survival mechanism that restores metabolic equilibrium and prevents further injury.”

Comment3 and 4: Finally, the Authors could elaborate more on pharmacologic induction of protective autophagy mentioning also off-target drugs potentially useful such as metformin and modulators of the adenosine A2A receptor. Linked to the above, the Authors could also discuss the potential of nutraceutical inducers of autophagy, such as for instance Resveratrol. See for instance: Totonchi, H., Mokarram, P., Karima, S. et al. Resveratrol promotes liver cell survival in mice liver-induced ischemia-reperfusion through unfolded protein response: a possible approach in liver transplantation. BMC Pharmacol Toxicol 23, 74 (2022). https://doi.org/10.1186/s40360-022-00611-4.

Response3 and 4: We have added the requested information based on the reviewer’s valuable comment:

“Line 368-374: Therapeutics like metformin [60] and adenosine A2A receptor [61] modulators augment autophagic flux and provide cytoprotection in models of liver damage.  Nutraceuticals such as resveratrol have exhibited the capacity to activate autophagy and mitigate oxidative stress, while also enhancing hepatocyte survival after I/RI via regulating the unfolded protein response [62].  These findings endorse the feasibility of repurposing established medicines to expedite the practical use of autophagy-based approaches in liver trans-plantation.”

Comment5: Please, spell out abbreviations at first use (e.g. line 34 I/RI)

Response5: All the abbreviations double checked and revised.

Comment6: A cartoon illustrating the pathways involved in autophagy induction and transplantation protection could help readers to catch up with the message.

Response6: We have added figure 2 to show the role of autophagy in key stages of Liver transplantation.

Round 2

Reviewer 1 Report

Comments and Suggestions for Authors

Note has been made of the revisions/alterations to the manuscript in response to the previous comments made by the reviewer

Reviewer 2 Report

Comments and Suggestions for Authors

Thanks for addressing all the issues in your revised manuscript, which I endorse for publication.